# Meta-sketch: A Neural Data Structure for Estimating Item Frequencies of Data Streams

## Abstract

To estimate item frequencies of data streams with limited space, sketches are widely used in real applications, including real-time web analytics, network monitoring, and self-driving. Sketches can be viewed as a model which maps the identifier of a stream item to the corresponding frequency domain. Starting from the premise, we envision a neural data structure, which we term the *meta-sketch*, to go beyond the basic structure of conventional sketches. The meta-sketch learns basic sketching abilities from meta-tasks constituted with synthetic datasets following *Zipf* distributions in the pre-training phase, and can be fast adapted to real (skewed) distributions in the adaption phase. Extensive experiments demonstrate the performance gains of the meta-sketch and offer insights into our proposals.

## 1 Introduction

Estimating item frequency is a basic topic in data stream processing, which finds applications in the fields of networking, databases, and machine learning, such as real-time data analyzing [1–4], network traffic monitoring [5–7], natural language processing [8] and search ranking [9]. Towards infinite data streams, a common class of solutions [10–15] use a compact structure taking sublinear space for counting the number of occurrences of each stream item, called the *sketch*.

Under the prevalent evidence of skewed distributions in data streams, *basic sketches* achieve the space compactness by hashing and approximately aggregating stream items. Basic sketches, including CM-sketch [10], C-sketch [11] and CU-sketch [12], use a 2D array of counters as the core structure. To optimize the sketching performance, there arise *augmented sketches* [13,14], which attach filters to basic sketches, to capture the preliminary patterns of skewed distributions (e.g., high/low-frequency items). By separately maintaining the filtered high/low-frequency items, augmented sketches strive to eliminate the estimation error incurred by hash collisions between the high- and low-frequency items. Further, *learned augmented sketches* [15] improve the filters of the augmented sketches by memorizing short-term high/low-frequency items via a pre-trained neural network (NN in short) classifier. But it is not clear how the pre-trained NN can be adapted to dynamic streaming scenarios, where the correspondence between items and frequencies varies. In a nutshell, sketches are structures compactly summarizing stream distributions to count item frequencies with limited space budgets.

From the retrospective analysis of sketches, an observation can be drawn that the evolution of sketches conforms with the exploitation of data distributions. It is thus a natural evolution to consider a sketch that generally and automatically captures more distribution patterns with limited space budgets. In this paper, we envision a novel neural sketch, called the *meta-sketch*, with techniques of meta-learning and memory-augmented neural networks. The meta-sketch learns the sketching abilities from automatically generated meta-tasks. Depending on the types of meta-tasks, we study two versions of the meta-sketch, called *basic* and *advanced meta-sketches*.

Submitted to 36th Conference on Neural Information Processing Systems (NeurIPS 2022). Do not distribute.

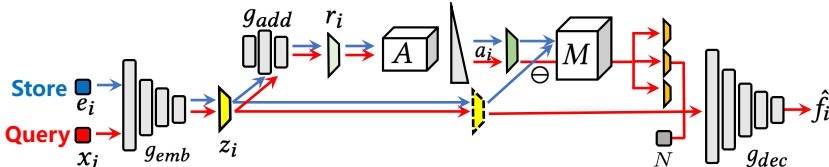

Figure 1: The Framework of the Meta-sketch

The basic meta-sketch implements the simulation of basic sketches, through the training process with basic meta-tasks following *Zipf* distributions, which are prevalent in the scenes of real data streams [16– 20]. The advanced meta-sketch extends the basic version to fast adapt to the specific runtime of stream processing, through the training with adaptive meta-tasks, which are generated by online sampling of real data streams. Our work follows a typical setting where the distribution of item frequencies follows a skewed distribution, but the correspondence between items and frequencies varies. For example, in software-defined networks (SDN), sketches are deployed to programmable switches to collect per-flow statistics, where IP packets follow *heavy-tailed* distributions [15, 21]. In distributed databases, it gives advances to collect statistics of data shards to optimize data placement and query caching, where query phrases follow approximate *Zipf* distributions [15]. Given that the item population follows a specific distribution, the local distributions, i.e., item-frequency correspondences on shards or flows, are different. Instead of retraining learned augmented sketches on each local distribution, the advanced-sketch can be quickly adapted to different local distributions once trained.

As a member of the neural data structure family [15, 22–24], the meta-sketch significantly differs from conventional sketches, in terms of the structure and working mechanism. The meta-sketch utilizes NN's powerful encoding/decoding capabilities to perceive data distributions and express and compress explicit or implicit information to retrieve item frequencies with better accuracies. Meanwhile, the meta-sketch is differentiable to fully perceive frequency patterns for self-optimization.

Our contributions are as follows. **1)** We propose the meta-sketch, the first neural data structure for the problem of item frequency estimation, based on meta-learning. **2)** The basic meta-sketch acquires sketching abilities by learning from synthetic datasets, and outperforms basic sketches in real datasets. The advanced meta-sketch automatically encompasses the ability analogous to the auxiliary structures deliberately devised in (learned) augmented sketches, yet yielding better accuracies and robustness when adapted to dynamic scenes. **3)** Through extensive empirical studies on real and synthetic datasets, we evaluate our proposed meta-sketches and analyze the mechanism of major modules.

## 2 Meta-sketch Structure

### 2.1 Preliminaries

We consider a standard data stream scenario [19]. Suppose a data stream $\mathcal{S}_N : \{e_1, ..., e_N\}$ with $N$ items and $n$ distinct items. Each item $e_i \in \mathcal{S}_N$ takes a value from the item domain $\mathbb{X} = \{x_1, ..., x_n\}$ where $x_i \neq x_j$. The frequency $f_i$ is equal to the number of times that item $x_i$ appears in $\mathcal{S}_N$.

To leverage learning techniques for item frequency estimation, a naïve way is to train a NN model (e.g., MLP/LSTM) that learns/memorizes the mapping relationship between items and frequencies with multiple training iterations, similar to [15, 22, 24]. However, it violates the typical setting of stream processing where item observations are transient and are therefore handled in one pass [18]. More, the costly procedure has to be repeated from the scratch for a new data stream. Inspired by the meta-bloom filter [23], we consider a case of one-shot learning (fitting for one-pass stream processing) by using meta-learning [25, 26] and memory-augmented networks [27, 28]. Meta-learning employs sampled meta-tasks to learn the ability to solve a class of domain tasks rather than memorizing patterns for a specific task. The memory-augmented networks incorporate external memories into NN models, significantly enhancing the potentials of NN models with more learnable parameters. Meanwhile, it performs efficient and explicit operations (i.e., reading and storing) for external memories, allowing NN models to process information similarly to conventional data structures.

The framework of the meta-sketch consists of $4$ functional modules, *Embedding* ($\mathcal{F}_E$), *Sparse addressing* ($\mathcal{F}_{Sa}$), *Compressed storage* matrix ($M$), and *Decoding* ($\mathcal{F}_{dec}$), as shown in Figure 1. Like traditional sketches, the meta-sketch encodes and memorizes online stream items in one pass, and answers queries by decoding corresponding item-frequency information from the structure.

Thus, we define $2$ operations, *Store* and *Query*. Specifically, the *Store* operation first passes each incoming stream item to $\mathcal{F}_E$ for the embedding representation, and then writes the embedding vector into $M$, according to the address derived by $\mathcal{F}_{Sa}$. When estimating the frequency of an item, the

*Query* operation calculates the item's address in $M$ via $\mathcal{F}_{Sa}$, reads the corresponding information vector from $M$, and decodes the item frequency by $\mathcal{F}_{dec}$ from the retrieved information vector .

## 2.2 Modules

**Embedding.** The module $\mathcal{F}_E$ has two purposes: **1)** performing representational transformation for an incoming item $e_i$ and mapping it into a dense embedding vector $z_i$ that holds implicit features about item-frequency distributions and serves as the basis for identifying stream items; **2)** decoupling the embedding vector $z_i$ to obtain a refined vector $r_i$, which is used to derive the address for reading/writing on the compressed storage matrix $M$.

Accordingly, $\mathcal{F}_E$ consists of the embedding network $g_{emb}$ and the address network $g_{add}$. We assume that an item $e_i \in \mathcal{S}_N$ is numerically encoded for the unique identification, following the conventions of stream processing [18, 19]. Thus, we have $z_i, r_i \leftarrow \mathcal{F}_E(e_i)$, where $z_i \leftarrow g_{emb}(e_i)$ and $r_i \leftarrow g_{add}(z_i)$. Here, $z_i \in \mathbb{R}^{l_z}$ is an embedding vector of length $l_z$, and $r_i \in \mathbb{R}^{l_r}$ is a refined vector of length $l_r$. The vector $z_i$ serves multiple intents: **1)** it makes a basis for deriving the address of an item in $\mathcal{F}_{Sa}$; **2)** it serves as the compressed vector of an item written into $M$; **3)** it works as a partial input of $\mathcal{F}_{dec}$ for decoding the item frequency; **4)** it also plays the role of perceiving/compressing patterns of a specific frequency distribution, as discussed in Section 5. In addition, to enhance the addressing functionality and eliminate other interference factors, we decouple $z_i$ to generate a refined vector $r_i$, instead of using $z_i$ directly for the addressing.

**Sparse addressing.** The module $\mathcal{F}_{Sa}$ aims to derive the address $a_i$ for storing the embedding vector $z_i$ into the storage matrix: $a_i \leftarrow \mathcal{F}_{Sa}(r_i)$. In terms of functionality, $\mathcal{F}_{Sa}$ is analogous to the hash functions of traditional sketches, except that $\mathcal{F}_{Sa}$ is parameterized and differentiable. Specifically, the addressing of the meta-sketch is done via a 3D addressing matrix $A$ of parameters to be learned and a sparse SoftMax function: $a_i \leftarrow SparseMax(r_i^T A)$, where $A \in \mathbb{R}^{d_1 \times l_r \times d_2}$. Then, the batch matrix multiplication of $A$ and the transpose of $r_i$ results in the addressing vector $a_i \in \mathbb{R}^{d_1 \times 1 \times d_2}$.

The setting of $d_1$ and $d_2$ determines the size of address space for storing the embedding vectors. Typical addressing methods [23, 28] use a 2D matrix ($l_r \times d_2$) for recording the mapping of an embedding vector to a slot ($d_2$ is the number of slots). In contrast, we add one more dimension $d_1$ to simulate the multi-hash setting of traditional sketches, in view of that a 2D addressing matrix can reach a differentiable simulation of a hash function [23, 24]. Matrix $A$ simulates multiple hash functions, yielding robust frequency decoding and the rationality of the learning optimization. Note that each 2D slice $A^*$ of $A$ is stacked from $d_2$-unit vectors $b_i \in \mathbb{R}^{l_r}$ by normalizing the parameters of $A$ at each gradient update of the training process. Normalized $A$ can avoid overflowing when compressing its size by reducing data precisions and enhance the interpretability (see Section 5).

In addition, we utilize sparse SoftMax [29, 30] instead of SoftMax to normalize the address $a_i$. It brings the following benefits by constraining some bits of $a_i$ to zero, which **1)** promotes quick derivation during the back-propagation; **2)** reduces the overhead of storage matrix accessing by skipping the slots of $M$ corresponding to the "0" bits of $a_i$; **3)** leads to de-noising with the vector compression.

**Compressed storage matrix.** We use a matrix $M \in \mathbb{R}^{d_1 \times l_z \times d_2}$ [1] to store an embedding vector $z_i \in \mathbb{R}^{l_z}$ in accordance to its address $a_i \in \mathbb{R}^{d_1 \times 1 \times d_2}$. The functionality of $M$ is similar to the 2D array of counters in traditional sketches, yet yielding better capabilities in the storage compression. Traditional sketches store item counts. Differently, $M$ stores embedding vectors, which have richer information compression capabilities, due to the diversity of value changes on different bits.

**Decoding.** Given a query item $x_i$, the module $\mathcal{F}_{dec}$, consisting of one NN component $g_{dec}$, decodes the information corresponding to $x_i$, in order to obtain the estimated frequency $\hat{f}_i$. The vector fed into $g_{dec}$ is the concatenation of vector $\{M \ominus a_i\}$, vector $z_i$, and the current number of items (i.e., $N$) recorded in a counter, $\hat{f}_i \leftarrow g_{dec}(\{M \ominus a_i\}, z_i, N)$. The operator $\ominus$ refers to the reading operation for the storage matrix. The basic form of $\ominus$ gives the operation as $M \ominus a_i = Ma_i^T$ [2] [27, 28]. For optimization, we consider two optimized forms of $\ominus$, inspired by the "count-min" mechanism of the CM-sketch. The first one gives the minimum value of each row in $Ma_i^T$, aiming to remove the noise of other items. The second one gives the minimum value of each row in $Ma_i^T \circ \frac{1}{z_i}$, a normalized

---

[1]In this paper, we control $l_r : l_z \approx 1 : 5$ to compress $A$.

[2]$a_i^T$ means transpose operation for dim 1 and $d_2$

136 form of $Ma_i^T$. Here, $\circ$ denotes the Hadamard product, and $z_i$ requires broadcast operations to comply
137 with its requirements. So, $\{M \ominus a_i\}$ refers to the concatenation of vectors generated by the basic
138 form and the two optimized forms. Please refer to supplement materials for more details.

## 2.3 Operations

140 **Operation Store** is performed by feeding an incoming item $e_i$ to $\mathcal{F}_E$ and $\mathcal{F}_{Sa}$ to obtain embedding
141 vector $z_i$ and address $a_i$, and then additively writing $z_i$ to $M$, weighted by $a_i$: $M \leftarrow M + z_i a_i$. Here,
142 other writing types [23, 26–28] can also be employed, but simple additive writing is more efficient
143 and allows to compute gradients in parallel [23]. In addition, additive writing also allows to define an
144 optional $Delete$ operation for the meta-sketch (see the supplement materials).

145 **Operation Query** estimates the frequency of a given query item $x_i$. First, $z_i$ and $a_i$ are obtained,
146 similar to that of operation Store. Then, the vectors $\{M \ominus a_i\}$ are retrieved from $M$ and $N$ can be
147 easily obtained by a small counter. Finally, $\{M \ominus a_i\}$, $z_i$ and $N$ are jointly fed into $g_{dec}$ to get the
148 estimated frequency $\hat{f}_i$ of $x_i$ as the returned value. The two operations are shown in Algorithm 1.

| **Algorithm 1:** Operations | **Algorithm 2:** Training Framework |
|---|---|
| 1 **Operation** Store($e_i$, $M$): | **Data:** Meta-sketch with all learnable parameters $\theta$, Meta-task sampler $R$; |
| 2 $\quad z_i, r_i \leftarrow \mathcal{F}_E(e_i)$ ; | 1 **while** $i$ *not reach max training steps* **do** |
| 3 $\quad a_i \leftarrow \mathcal{F}_{Sa}(r_i)$; | 2 $\quad$ Sample a meta-task $t_i : \{s_i, q_i\} \sim R$ and count $N$; |
| 4 $\quad M \leftarrow M + z_i a_i$; | 3 $\quad$ **for** $e_j^{(i)} \in s_i$ **do** Store($e_j^{(i)}$, $M$); **end** |
| 5 **Operation** Query($x_i$,$M$,$N$): | 4 $\quad$ **for** $x_j^{(i)}, f_j^{(i)} \in q_i$ **do** $\hat{f}_j^{(i)} \leftarrow$ Query($x_j^{(i)}$, $M$,$N$); $\mathcal{L} \mathrel{+}=$ LossFun($f_j^{(i)}, \hat{f}_j^{(i)}$); |
| 6 $\quad z_i, r_i \leftarrow \mathcal{F}_E(x_i)$; | 5 $\quad$ Backprop through: $d\mathcal{L}/d\theta$ and update parameters: $\theta \leftarrow$ Optimizer($\theta, d\mathcal{L}/d\theta$); |
| 7 $\quad a_i \leftarrow \mathcal{F}_{Sa}(r_i)$; | 6 $\quad$ Normalize A; |
| 8 $\quad \hat{f}_i \leftarrow \mathcal{F}_{dec}(\{M \ominus a_i\}, z_i, N)$; | 7 $\quad$ Clear $M$; |
| 9 $\quad$ **return** $\hat{f}_i$; | 8 **end** |

## 3 Meta-sketch training

### 3.1 Training Framework

152 The meta-sketch employs an efficient one-shot meta-training method [31]. The training process thus
153 contains two phases, *pre-training* and *adaption* phases. In the pre-training phase, the meta-sketch
154 learns an initial set of module parameters, including $g_{emb}$, $g_{add}$, $A$, and $g_{dec}$. The pre-training
155 goes offline across training units, i.e., basic meta-tasks, to acquire the ability of stream frequency
156 estimation. Then, in the adaption phase, the pre-trained meta-sketch goes fast across a set of light-
157 weighted training units, i.e., adaptive meta-tasks, to quickly acquire the task-specific knowledge, i.e.,
158 parameters for sketching real data streams at runtime.

159 The training units, i.e., meta-tasks, are crucial for both phases. The training process of the meta-sketch
160 on a single meta-task is equivalent to simulating storing and querying an instance of data streams
161 while computing the estimation error to optimize the learnable parameters. Thus, a meta-task $t_i$
162 consists of a store set $s_i$ (also called a support set) and a query set $q_i$. The store set $s_i$ can be viewed
163 as an instance of data streams, $s_i : \{e_1^{(i)}, ..., e_{N_i}^{(i)}\}$, where $N_i$ is the number of stream items in $s_i$. The
164 query set $q_i$ can be represented by a set of items from the stream instance with paired frequencies in
165 the store set $s_i$, formally, $q_i : \{(x_1^{(i)} : f_1^{(i)}), ..., (x_{n_i}^{(i)} : f_{n_i}^{(i)})\}$, where $n_i$ is the number of distinct items
166 in $s_i$. In this work, we define two types of meta-tasks, *basic* (Section 3.2) and *adaptive* (Section 3.3)
167 meta-tasks, corresponding to the pre-training and adaption phases, respectively.

168 The two training phases, that are based on different types of meta-tasks, follow the same training
169 framework, as shown in Algorithm 2, except for the sampler and initial parameters. To optimize on
170 reducing both absolute and relative frequency estimation errors[3], we devise an adaptive hybrid loss
171 function [32] for the meta-sketch: $\frac{1}{2\sigma_1^2}(f_i - \hat{f}_i)^2 + \frac{1}{2\sigma_2^2}|f_i - \hat{f}_i|/f_i + log\sigma_1\sigma_2$, where $\sigma_1$ and $\sigma_2$ are
172 learned parameters, and $f_i$ and $\hat{f}_i$ are the true and estimated frequencies of item $x_i$, respectively.

---

[3]Average Absolute Error: $AAE = \frac{1}{n}\sum_{i=1}^n |f_i - \hat{f}_i|$; Average Relative Error: $ARE = \frac{1}{n}\sum_{i=1}^n \frac{|f_i-\hat{f}_i|}{f_i}$.

## 3.2 Basic Meta-task Generation

In the pre-training phase, basic meta-tasks should make the meta-sketch to simulate traditional sketches and preserve certain generality without relying too much on the patterns of specific distributions (Section 5). Therefore, we generate meta-tasks based on the Zipf distribution, which is found to be prevalent in real scenes of data streams [16–20].

A meta-task is essentially a data stream instance with item size $n$, which can be determined by the total number of items $N$ and the relative frequency distribution $p$. Alternatively, we can generate meta-tasks by presupposing different $n$, $\bar{f}$ and $p$, where $\bar{f}$ is the frequency mean, since $N = \bar{f} \times n$. Thus, basic meta-task generation is based on a sampler $R : \{I, L, P\}$, as follows.

An **item pool** $I$ is a subset of the item domain $\mathbb{X}$. The cardinality of $I$ is in relevance to the identification capability of the meta-sketch. If the item domain is known a-priori, it can be directly taken as the item pool. Otherwise, in applications where the item domain is only partially known or even unknown, the item pool can be constructed by sampling from the historical records. Even in the case that the item pool does not completely cover the item domain, the "missing" item can still be identified, due to the homogeneity of the domain-specific embedding space, given that the number of distinct items does not meet the item pool capacity $|I|$.

A **frequency mean range** $L$ is the range for the frequency mean $\bar{f}$. One can get the value of $\bar{f}$ by statistics of each sampled stream instance and extract the minimum and maximum $\bar{f}$s to build $L$.

A **distribution pool** $P$ consists of many instances generated according to different parameters of relative frequency distributions. In this paper, we consider a family of *Zipf* distributions [33] with varied parameter $\alpha$, as the base for constructing $P$. $\alpha$ can be selected from a wide range to have a good coverage of different distributions.

Notice that the meta-tasks are for the meta-sketch to learn the sketching ability, instead of spoon-feeding the meta-sketch to mechanically memorize the parameters of $R$. It means that the trained meta-sketch has the generalization ability to handle the case not covered in $R$ (see Section 4.2).

The generation of a meta-task $t_i$ can be done based on sampler $R$, as follows. We first randomly sample a subset of $n_i$ items from $I$, and a frequency mean $\bar{f}_i \in L$. Then, we sample a distribution instance $p_i \in P$ and make the $n_i$ items' frequencies conform to $p_i$ and $\bar{f}_i$. For example, the frequencies of $n_i$ items can be set as $n_i \times \bar{f}_i \times p_i$, where $p_i \sim Zipf(\alpha)$ is a random variable. The above steps are repeated until the store set $s_i$ and query set $q_i$ are built.

## 3.3 Adaptive Meta-task Generation

While processing real data streams, we can get the item set $I_r$ and its distribution $p_r$ by online sampling. $I_r$ and $p_r$ are then used for generating the set of adaptive meta-tasks. For each adaptive meta-task, an item subset is sampled from $I_r$, and the relative frequency corresponding to each item is sampled from $p_r$. The process is similar to the generation of basic meta-tasks. The only difference from basic meta-task generation is that, there is no distribution pool anymore, because the real data stream is unique. Also, we intentionally randomize the correspondence between an item and its real relative frequency on the original data records. It is equivalent to constructing meta-tasks where the item frequencies dynamically change. For example, the frequency of an item may first increase, then suddenly drop [21]. With adaptive meta-tasks, the meta-sketch learns to quickly adapt to the distribution $p_r$, while being flexible against the item frequency change. The detailed algorithms of generating basic/adaptive meta-tasks are shown in supplement materials.

# 4 Experiments

## 4.1 Basic Setup

**Dataset.** We use two real datasets. *Word-query* is a streaming record of search queries, where each query contains multiple words (e.g., "News today") [15]. *IP-trace* consists of IP packets, where each packet is identified by a unique source/destination address pair (e.g., 192.168.1.1/12.13.41.4) [21]. We assume that query phrases and IP addresses are numerically encoded, similar to [15].

Table 1: Results of Basic Meta-sketch ($T_r$)

| Method | Metrics | Word-query | | | | IP-trace | | | |
|---|---|---|---|---|---|---|---|---|---|
| | | n=5K, B=9KB | n=10K, B=11KB | n=20K, B=13KB | n=40K, B=15KB | n=5K, B=9KB | n=10K, B=11KB | n=20K, B=13KB | n=40K, B=15KB |
| Basic MS | ARE | **12.3** | **14.74** | **10.98** | **13.79** | **3.00** | **1.51** | **2.97** | **1.13** |
| | AAE | **31.54** | **38.54** | **40.63** | **53.67** | **5.57** | **5.01** | **6.94** | **5.56** |
| CS | ARE | 32.94 | 57.97 | 98.01 | 162.43 | 6.08 | 9.94 | 15.57 | 24.49 |
| | AAE | 57.54 | 101.44 | 172.44 | 282.59 | 10.42 | 16.82 | 26.46 | 41.91 |
| CMS | ARE | 21.34 | 48.33 | 111.82 | 239.11 | 8.12 | 16.07 | 32.77 | 65.19 |
| | AAE | 38.04 | 84.62 | 195.61 | 416.01 | 13.67 | 27.39 | 55.29 | 110.65 |

Table 2: Results of Basic Meta-sketch ($T_s$)

| Method | Metrics | n=5K,B=9KB | | | n=10K,B=11KB | | | n=20K,B=13KB | | | n=40K,B=15KB | | |
|---|---|---|---|---|---|---|---|---|---|---|---|---|---|
| | | 0.5 | 1.1 | 1.5 | 0.5 | 1.1 | 1.5 | 0.5 | 1.1 | 1.5 | 0.5 | 1.1 | 1.5 |
| Basic MS | ARE | **0.43** | **1.05** | **2.63** | **0.73** | **3.25** | **3.14** | **0.47** | **1.67** | **1.35** | **0.43** | **2.58** | **9.65** |
| (Word-query) | AAE | **24.7** | **17.72** | **8.93** | **31.24** | **27.02** | **9.41** | **27.29** | **22.19** | **9.2** | **25.04** | **26.95** | **19.87** |
| Basic MS | ARE | **0.59** | **2.27** | **9.38** | **0.73** | **0.86** | **1.02** | **0.72** | **1.73** | **7.52** | **0.73** | **0.79** | **2.33** |
| (IP-trace) | AAE | **26.45** | **21.49** | **14.73** | **38.33** | **19.32** | **7.95** | **35.48** | **22.28** | **15.74** | **39.57** | **21.75** | **14.06** |
| CS | ARE | 1.98 | 6.72 | 10.99 | 2.7 | 12.12 | 16.9 | 3.73 | 20.8 | 27.46 | 5.17 | 37.96 | 43.76 |
| | AAE | 74.96 | 47.98 | 15.89 | 102.05 | 75.83 | 23.8 | 140.65 | 118.29 | 38.7 | 194.32 | 198.4 | 59.96 |
| CMS | ARE | 4.96 | 7.52 | 5.47 | 9.27 | 15.85 | 9.44 | 17.29 | 32.7 | 16.38 | 32.24 | 66.35 | 27.89 |
| | AAE | 187.52 | 53.81 | 8.17 | 350.08 | 99.82 | 13.58 | 651.63 | 185.54 | 22.88 | 1213.38 | 347.32 | 38.18 |

**Baseline.** We hereby evaluate the basic and advanced meta-sketches. From now on, we use MS to represent the term meta-sketch for brevity. We compare basic MS (after the pre-training phase) with CM-sketch (CMS) and C-sketch (CS). We compare the advanced MS (after the adaptation phase) with learned augmented sketch (LS) and cold filter (CF), which are two variants of CM/C sketches with auxiliary structures. According to the default setting [10, 11], the number of hash functions for all sketches is 3. We adopt two commonly accepted metrics for evaluating the accuracies of stream frequency estimation, AAE and ARE[3].

**Parameters.** We implement $g_{emb}$ or $g_{add}$ in MLP with 2-layers of sizes 128 and 48, followed by batch normalization, and $g_{dec}$ in an MLP with 3-layers of 256 with residual connections. We use the $relu$ function for layer connections. The space budget $B$ is spent on storing $M$, the same as the setting in neural data structures [23]. Other modules, like hashing libraries, are commonly accepted as reusable and amortizable resources for multi-deployment of sketches [21, 23]. Note that due to space limitations, the details and methods of parameter settings of $M(A)$, the ablation experiments and some parameter discussions are shown in the supporting material.

## 4.2 Basic Meta-sketch

**Settings.** For each dataset, we train the basic MSs under 4 item pools with $\{5K, 10K, 20K, 40K\}$ different items, respectively. The meta-task sampler are with *Zipf* distributions. We build the distribution pools set with $\alpha \in [0.8, 1.3]$ and set frequency mean range $L = [50, 500]$. For basic meta-sketch training, the default maximum number of training steps $\phi$ is 5 million, the learning rate is 0.0001, and the $Adam$ optimizer is used. For evaluation, we consider two types of tasks, $T_r$ and $T_s$. $T_r$ are directly obtained by random sampling on two real data streams with different values of $n$, i.e., the number of distinct items. Note that the frequency distributions of $T_r$ are not necessarily obey *Zipf* distributions. $T_s$ are the synthetic tasks, where the item frequency follows the *Zipf* distribution with $\alpha \in \{0.5, 1.1, 1.5\}$. To evaluate the generability and stability of basic MS, both $T_s(0.5)$ and $T_s(1.5)$'s distributions are not covered by the distribution pool of the meta-task samplers.

**Performance.** Table 1 shows the performance of all competitors based on real dataset $T_r$. It shows that the basic MS outperforms traditional basic sketches, i.e., CMS and CS, on all testing cases. For example,the results on IP-trace show that, when $n$=40K and $B$=15KB, the ARE of basic MS is 1.13, while AREs of CMS and CS are 65.19 and 24.49, respectively. The advantage of meta-sketch is significant when testing on $T_s$ with different $\alpha$s, as shown in Table 2. Note that we use random choices to simulate the ideal hash functions for traditional sketches like [15], so that CS and CMS have the same result on test tasks with the same $\alpha$ in both datasets.

We show the trend of ARE w.r.t. the space budget, in Figure 2 ($T_r$, $n$=5K, Word-query). Compared to the dramatic performance degrading of traditional sketches, basic MS holds stable performance. We show that the trend of ARE w.r.t. the number of distinct items in Figure 3 ($T_r$, $B$=9KB, Word-query). Compared to traditional sketches, the ARE of basic MS increases sub-linearly w.r.t. the value of $n$. Note that AAE has similar results for the above experiments, see the supplement materials.

**Generalization.** We test the generality of basic MS to new items that are not in the item pool of the meta-task sampler in Figure 4(a). We make the experiments ($n$=5K, $B$=9KB, Word-query) by replacing some items in $T_r$ with new items, and vary the fraction of new items to observe the trend of the performance. It shows that the ARE/AAE moderately increases w.r.t. the ratio

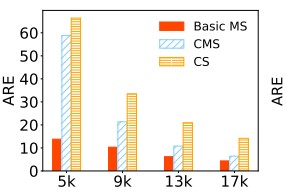

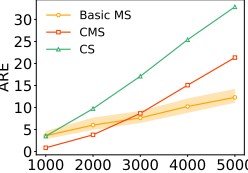

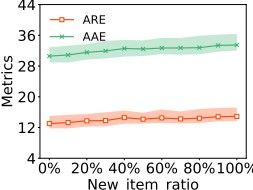

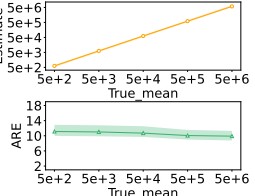

(a) New Items      (b) New frequency means

Figure 2: ARE w.r.t. $B$    Figure 3: ARE w.r.t. $n$     Figure 4: Generality of Meta-sketch

Table 3: Results of Advanced Meta-sketch

| Method | Metrics | Word-query | | | | IP-trace | | | |
|---|---|---|---|---|---|---|---|---|---|
| | | n=5K B=9KB | n=10K, B=11KB | n=20K B=13KB | n=40K B=15KB | n=5K B=9KB | n=10K B=11KB | n=20K B=13KB | n=40K B=15KB |
| Advanced | ARE | **3.05** | **2.83** | **4.06** | **5.20** | 0.87 | **0.89** | **1.38** | **2.29** |
| MS | AAE | 21.42 | **26.11** | **35.00** | **43.81** | 3.77 | 4.46 | **5.13** | **6.55** |
| CF90 | ARE | 3.58 | 14.53 | 141.70 | 1127.11 | **0.85** | 2.74 | 4.20 | 16.71 |
| | AAE | **21.13** | 59.18 | 381.63 | 2217.28 | **1.32** | **3.01** | 7.71 | 31.20 |
| CF70 | ARE | 7.95 | 29.02 | 139.87 | 541.37 | 1.51 | 3.10 | 8.95 | 46.79 |
| | AAE | 29.02 | 76.58 | 295.63 | 970.94 | 2.57 | 5.51 | 16.83 | 82.84 |
| CF40 | ARE | 91.16 | 138.64 | 244.24 | 407.83 | 12.62 | 33.50 | 103.76 | 155.61 |
| | AAE | 174.86 | 252.22 | 421.85 | 693.47 | 24.16 | 60.79 | 175.14 | 279.72 |
| LCMS(1%) | ARE | 20.52 | 48.69 | 111.85 | 266.50 | 8.34 | 17.09 | 35.22 | 77.79 |
| | AAE | 37.80 | 81.93 | 194.15 | 451.28 | 13.72 | 28.39 | 59.10 | 129.86 |
| LCS(1%) | ARE | 25.53 | 40.84 | 67.21 | 104.54 | 5.20 | 7.80 | 11.33 | 17.12 |
| | AAE | 44.53 | 78.17 | 122.57 | 180.56 | 8.78 | 13.10 | 18.97 | 28.38 |

of new items. The performance is acceptable considering the fact that the item domain is often stable in practical applications. We then test the generality of meta-sketches to varied frequency means that are not in range $L$ of the meta-task sampler, as shown in Figure 4(b). The experiment ($n$=5K, $B$=9KB, Word-query) is done by sampling a series of $T_s$ tasks with frequency means in $\{500, 5K, 50K, 500K, 5000K\}$. It shows that as the mean of the true frequencies increases, the estimated frequencies of the meta-sketch increase linearly, so that the ARE keeps stable.

### 4.3 Advanced Meta-sketch

**Settings.** The generation of adaptive meta-tasks is similar to that of basic meta-tasks (Section 3.2), except that each item pool reads real frequency distributions for the adaption as described in Section 3.3. In the adaption phase, the maximum number of training steps is $0.002 * \phi$.

**Performance.** Table 3 compares the performance of advanced MS with traditional sketches and their variants, LS and CF, on real dataset $T_r$. We implement two LSs according to [15], learned CM-sketch (LCMS) and learned C-sketch (LCS), following the default setting that (top 1%) high-frequency items are separately stored. For CF, we follow the parameter setting in [14], and use CF40, CF70, and CF90 for setting the filter percentages to 40%, 70%, and 90% of the total size, respectively. It shows that the advanced MS achieves a better performance than LSs and CFs. Also, AAE/ARE of advanced MS increases more moderately w.r.t. the number of distinct items $n$, compared to its competitors.

Furthermore, we compare the performance of the advanced MS and the LS under dynamic streaming scenarios, as shown in Figure 5. We select a set of $T_r$ ($n$=5K,$B$=9KB,Word-query), and gradually shuffle the correspondence between items and frequencies. Here, the shuffle ratio is increased from 0 to 100%. It shows that the average ARE of advanced MS only slightly fluctuates between 3.26 and 4.0, and the average AAE is in the range of 21.28 and 21.68. In contrast, AAE of LCS or LCMS starts above 37, and increase significantly w.r.t. the increase of the shuffle ratio. Actually, the classifier of LS tends to incur more errors due to the gradual shift of high- and low-frequency items, resulting in an increased number of hash collisions, thus deteriorating the estimation accuracy.

## 5 Analysis

The meta-sketch is trained based on meta-tasks, consisting of various stream distributions. We expected that the meta-sketch can learn the ability to sketch item frequencies. Somehow, it is unavoidable that the meta-sketch's ability is limited by patterns of given meta-tasks. Thus, setting up the two training phases benefits the balance of the trade-offs. In the pre-training phase, we select the most representative *Zipf* distribution to form basic meta-tasks, making the basic meta-sketch adaptable to a wide range of data streams. In the adaptation phase, we sample adaptive meta-tasks from raw data streams to make the advanced meta-sketch more specialized. Next, we analyze the

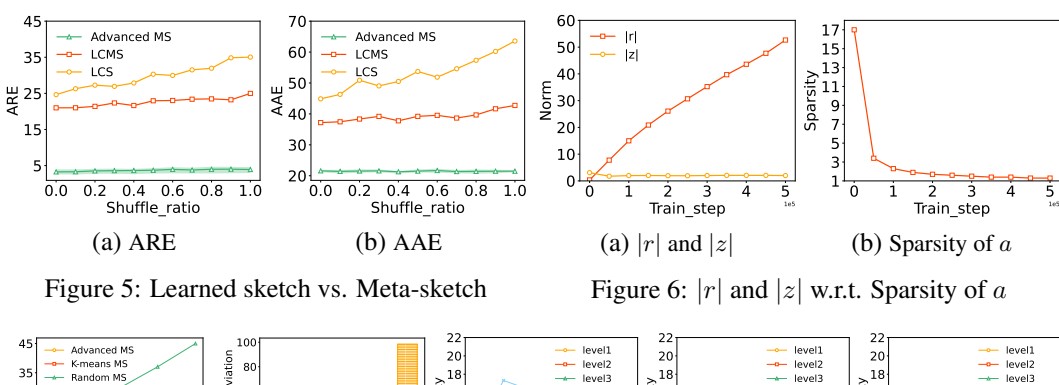

(a) ARE      (b) AAE      (a) $|r|$ and $|z|$      (b) Sparsity of $a$

Figure 5: Learned sketch vs. Meta-sketch      Figure 6: $|r|$ and $|z|$ w.r.t. Sparsity of $a$

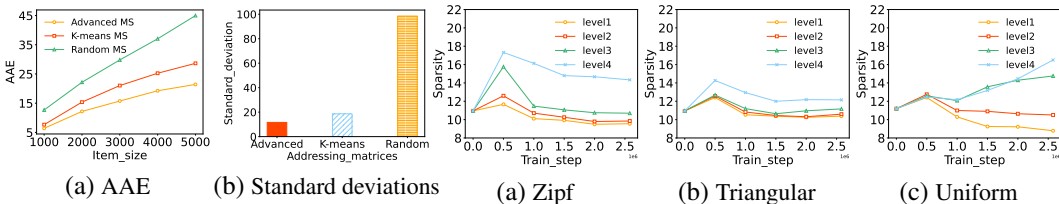

(a) AAE    (b) Standard deviations    (a) Zipf    (b) Triangular    (c) Uniform

Figure 7: Three addressing matrices      Figure 8: The sparsity of embedding vectors

working mechanism of the three modules of the meta-sketch as well as their roles in acquiring the two abilities.

**Sparse Addressing Module.** We take a 2D slice $A^*$ (size is $l_r \times d_2$) of the $A$ matrix to analyze the process of a refined vector $r$ getting addressing $a$ through this module. First, we have $a \leftarrow SparseMax(r^T A^*) \Rightarrow a \leftarrow SparseMax(\langle r \cdot b_1, r \cdot b_2, ..., r \cdot b_{d_2}\rangle)$. Since $b_i$ are unit vectors, we can get $a \leftarrow SparseMax(|r|c)$, $c = \langle cos\theta_1, cos\theta_2, ..., cos\theta_{d_2}\rangle$, where $\theta_i$ is the angle between $r$ and $b_i$. We continue to transform the form to get addressing $a \leftarrow Sparsegen(c; u; \frac{|r|-1}{|r|})$ [30], where $u$ is a component-wise transformation function applied on $c$. in this paper, we set $u(c){=}c$.

Based on the principle of $Sparsegen$ [30], $|r|$ mainly affects the sparsity (i.e., the proportion of non-zero bits in the vector) of $a$ during training process, while $c$ determines the positions and values of non-sparse bits. The Figure 6 shows a strong correlation between the average $|r|$ and the sparsity of $a$ during training from scratch ($n{=}5K$, $B{=}9KB$, Word-query, Basic MS). Since the embedding vector $z$ does not directly participate in the addressing process, the average $|z|$ remains stable. Further, we observe that the sparsity of $a$ will eventually converge to around 1, which means that each item is generally stored in a slot corresponding to the refined vector $r$ and the unit vector in $A^*$ with the maximum cosine similarity.

Therefore, the role of $A^*$ is to map refined vectors to the addressing vectors. The $d_2$ unit vectors in $A^*$ are the reference standard for mapping, which is equivalent to the mutually exclusive $d_2$-divisions of the refined vector space. Follow this point, we construct two matrices $K^*$ and $R^*$ of the same size as $A^*$. Among them, the $d_2$ unit vectors in $K^*$ come from the cluster centers of the sampled refined vectors. To achieve mutually exclusive division, we perform $Kmeans$ clustering with $K = d_2$ and *Cosine similarity* criterion. Then, we normalize the resulting $d_2$ cluster centers and stack them as $K^*$. In contrast, the unit vectors in $R^*$ are entirely randomly generated.

Figure 7 (a) shows the results of replacing $A^*$ on the trained meta-sketch with $K^*$ and $R^*$. The meta-sketch with $R^*$ shows the worst performance, but the performance of the meta-sketch with $K^*$ is close to the original $A^*$. Furthermore, We count the number of items mapped in every slot of $A^*$, $K^*$, $R^*$ and show their standard deviation in Figure 7 (b). The standard deviation of $R^*$ is much higher than $A^*$ and $K^*$, and a better meta-sketch tends to store items more evenly in each slot. Thus, The addressing module simulates the traditional sketch mechanism. Its principal function is to store the embedding vectors of items as evenly as possible in multiple memory slots, and an item is written to only one slot.

**Embedding Module.** The major source of conflicts in the meta-sketch is the stacking of different embedding vectors in a single slot. Thus, the sparsity of the embedding vector becomes an important indicator to determine the degree of conflicts. Figure 8 shows the relation between the sparsity of embedding vectors and the stream distributions ($n{=}5K$, $B{=}9KB$, Word-query, advanced MS). We select the meta-tasks under *Zipf*, *Triangular*, and *Uniform* distributions with different skewness levels (the definition of skewness and corresponding distribution parameters are shown in the supplement materials). The results show that the sparsity of the embedding vector is positively proportional to

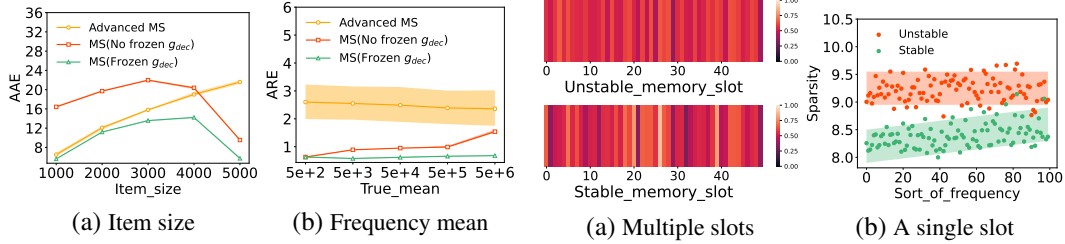

| (a) Item size | (b) Frequency mean | (a) Multiple slots | (b) A single slot |

Figure 9: Generality w.r.t. Decoding module

Figure 10: Unstable case vs. Stable case

the skewness of a distribution. Therefore, we speculate that the meta-sketch memorizes the pattern information of the distribution being adapted by self-tuning the sparsity of embedding vectors.

**Decoding Module.** The decoding module, as the deepest NNs in the meta-sketch, integrates various information to predict the item frequency and achieves generalization ability. To verify this, we adapt the advanced MS ($n$=5K, $B$=9KB, Word-query) to a special adaptive meta-task. The meta-task was sampled from the real data stream but with a fixed item size (5000) and frequency mean (250). Meanwhile, we do not change the correspondence between items and frequencies. Such meta-task forces the meta-sketch to pay more attention to the fixed patterns and thus limit its generalization.

Thus, we train the advanced MS with (or without) freezing the decoding module parameters based on the above meta-task. Figure 9 (a) shows the performance changes of the three models (advanced MS as baseline) on the evaluation tasks ($T_r$) of different item sizes. Without the frozen decoding module, the meta-sketch loses generalization ability at extended item sizes other than 5000. On the contrary, the meta-sketch with the frozen decoding module still retains the generalization ability and further utilizes the data stream pattern compared to the advanced MS, achieving the best performance. Similarly, as shown in Figure 9 (b), the meta-sketch without the frozen decoding module also loses a certain generalization ability in terms of frequency mean.

Actually, the above meta-task (termed as the *stable case*) can be viewed as a special case of an ordinary adaptive meta-task (termed as the *unstable case*). As a matter of fact, augmented sketches utilize frequency patterns similar to the stable case. For example, the learned augmented sketch memorizes (relatively) stable correspondence between items and frequencies, for filtering high-frequency items. To understand the meta-sketch's self-optimizing mechanism from the unstable case to the stable case, we analyze the storage of high/low-frequency items between multiple slots and a single slot in the memory. In Figure 10 (a), we show density heat-maps of low-frequency (below the top 20% high frequencies) items, stored by meta-sketches of stable and unstable cases on a 2D slice ($d_1$=2) of the storage matrix $M$, where the x-axis is the index of slots. The two heat-maps show that the meta-sketch under the stable case can store the low-frequency items concentratedly in some slots to avoid the conflicts with high-frequency items. Interestingly, the meta-sketch does not intentionally do this like augmented sketches. Instead, it is achieved by self-optimization during the training. Furthermore, Figure 10 (b) shows the relation between the sparsity of the embedding vector of items stored in a single slot and the frequency order, where the $x$-axis represents the frequencies in the ascending order. We speculate that the meta-sketch autonomously adjusts the sparsity of the embedding vector within a single slot in the stable case, so that the high/low-frequency items are automatically separated.

## 6 Conclusion

In this paper, we propose a neural data structure, called the meta-sketch, for estimating item frequencies in data streams. Unlike traditional sketches, the meta-sketch utilizes meta-learning and memory-augmented neural networks. The meta-sketch is pre-trained with *Zipf* distributions and can be fast adapted to specific runtime streams. We study a series of techniques for constructing the meta-sketch. We also devise the generation of basic and adaptive meta-tasks corresponding to the pre-training and adaption phases, respectively. Extensive empirical studies on real datasets are done to evaluate our proposals. In the future, it is interesting to extend our proposal to other sketching tasks that are supported by traditional sketches.

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
