# OpenReview forum: "Meta-sketch: A Neural Data Structure for Estimating Item Frequencies of Data Streams"
_NeurIPS.cc/2022/Conference — NeurIPS 2022 Submitted_

### Official Review · Reviewer_4vDr · 2022-07-09

**Rating:** 7
**Confidence:** 3
**Soundness:** 4 excellent
**Presentation:** 3 good
**Contribution:** 3 good

**Summary:**

The work considers the problem of estimating item frequencies in data streams. Sketches are probabilistic data structures that solve this problem approximately; e.g., a classical count-min (CM) sketch increments counts resulting from several hash functions for each query (exactly like a Bloom filter, only with integer counts) and outputs the minimal value of these counts as the result; this is a biased estimator that can overestimate but not underestimate. Machine learning models can be used in sketches to exploit the properties of underlying data distributions. For example, it would be better to assign heavy hitters separate buckets in a CM-sketch to avoid collisions, and a machine learning model can be used to identify these heavy hitters when they appear.

The work presents Meta-Sketch, a neural data structure that emulates a sketch in a neural network. The Meta-Sketch is a neural network with an encoder that produces embeddings $z_i$ and refined embeddings $r_i$ of incoming items, a sparse addressing unit that converts $r_i$ into an address $a_i$ in the storage matrix (this is the neural version of hash functions), a compressed storage matrix $M$ that stores embedding vectors, and a decoder that produces the frequency from a reading from the storage matrix, embedding vector $z_i$, and the total number of stored items $N$. The network supports two kinds of operations: "Store" that additively writes $z_i$ to $M$ and "Query" that estimates the frequency via the decoder.

The entire network is both pretrained and fine-tuned (adapted) on a series of meta-tasks; each meta-task is a set of store and query operations; during pretraining, the meta-tasks are synthetic problems generated based on the Zipf distribution (the result is called a basic meta-sketch), and during adaption the network is fine-tuned on real data records (advanced meta-sketch). The idea here is to pretrain the architecture to operate as a sketch while preserving the ability to quickly adapt to a given data distribution.

The authors evaluate on two real datasets, a stream of search queries (several words each) and an IP trace where packets are identified with source/destination IP addresses. They report improved performance over classical sketches and also over previous attempts of adding ML to sketches: learned CM-sketch and learned C-sketch. The paper also presents an analysis section that studies how different units of the architecture perform.

**Questions:**

1. There is an ablation study in the supplement but I think the paper would benefit from a discussion of neural network architecture choices. For example, are there gains to be achieved from more involved encoder/decoder architectures?

2. It appears that the authors simply picked the same real datasets that were used in previous work (references [15, 21]). Are there other use-cases and available datasets for evaluation? Are there differences in the data distributions between these two (it appears there should be: IP traces are generally more heavy-tailed) and does it make any difference to Meta-Sketch results?

**Limitations:**

Limitations of the work and possible extensions have been adequately discussed in the conclusion. The societal impact question is not applicable in this case.

**Strengths And Weaknesses:**

Strengths:
* novel neural data structure with a clear use case
* analysis section supporting that the components operate as intended
* convincing experimental evaluation

Weaknesses (see also questions):
* lack of discussion of the architectural choices for the neural networks
* only two real datasets with no motivation for this choice and no discussion of their differences

---

> ### Author Response · Authors · 2022-08-02
> **Response to Reviewer 4vDr**
>
> Thanks a lot for your comments! We respond to your comments below.
> >### Q1: Discussion of the architectural choices for the neural networks...
>
> Thanks for your suggestion. We agree that there is room for improving the network structure, which is not intentionally optimized in this work. Some clues can be drawn from the study for the meta-sketch mechanism in Section 5 for optimizing each module of meta-sketch. For example, we can use the pre-trained embedding module to speed up the training of the meta-sketch, and use more read heads to provide more information to the decoder. However, a complex network structure may not bring performance gains, due to the trade-off with the corresponding storage overhead.
>
> When designing the meta-sketch, we also thought about the encoder/decoder, due to the potential benefits: 1) The enhancement in extracting patterns of frequency distributions. 2) The flexibility in supporting data items of different lengths. Meanwhile, there are some technical challenges: 1) The encoder/decoder architecture increases the complexity of meta-training, which requires updating the gradients in two stages, making the training of encoder/decoder unstable. 2) The encoder/decoder architecture may increase the write/query latency of the meta-sketch, because the network is bigger and deeper.
>
>  Therefore, we choose the current solution for the meta-sketch, which is simple and effective according to the empirical studies. We will continue to explore the possibility of incorporating encoder/decoder in future.
>
>
>
>
> >### Q2: Two real datasets with no motivation for this choice and no discussion of their differences...
>
> For fair comparison, we choose the datasets of [15,21], Word-query and IP-trace, to be in line with the competitor. IP-trace follows heavy-tailed distributions [15, 21] and the word-query follows Zipfian distributions [15]. The basic meta-sketch is trained with automatically and synthetically generated meta-tasks, so that there is no difference except for the domain of the data items if real datasets are changed. For the advanced meta-sketch, the frequency distributions of real datasets are captured by generating adaptive meta-tasks through online sampling.
>
> We further run experiments on two more datasets (Webdocs and Kosarak in http://fimi.uantwerpen.be/data/ ), which are commonly used for evaluating conventional sketches. In Webdocs and Kosarak, items are represented by numerically encoded IDs and frequency distributions are all approximate Zipf distributions. We use the 16-bit binary encoding of the ID as the input for the embedding module. The results are shown as follows. On both datasets, the advanced meta-sketch outperforms its competitors, which is consistent with the results on Word-query and IP-trace. Note that the performance of CF90 on Kosarak is close to that of the advance meta-sketch, because $n$ and $N$ is small. The advantage of meta-sketch is more significant on larger data streams (with a bigger $n$ or $N$), as shown in Table 3.
>
> | Webdocs (n=5K,B=9KB,N=152289) |   ARE  |   AAE  |
> |:--------------------:|:------:|:------:|
> |          CMS         |  7.79  | 10.52  |
> |          CS          | 10.45  | 14.70  |
> |       **Basic MS**       |  3.18  |  7.19  |
> |         CF90         |  0.71  | 21.23  |
> |         CF70         |  1.25  | 22.02  |
> |         CF40         | 11.21  | 37.91  |
> |       LCMS(1%)       |  7.28  |  9.55  |
> |        LCS(1%)       |  6.48  |  8.37  |
> |      **Advanced MS**     |  1.23  |  4.92  |
>
> | Kosarak (n=5K,B=9KB,N=79371) |   ARE  |   AAE   |
> |:--------------------:|:------:|:-------:|
> |          CMS         | 12.77  |  26.11  |
> |          CS          | 11.42  |  23.44  |
> |       **Basic MS**       |  3.89  |  12.58  |
> |         CF90         |  1.59  |  7.24   |
> |         CF70         |  3.26  |  12.33  |
> |         CF40         | 52.63  | 118.98  |
> |       LCMS(1%)       | 12.29  |  25.17  |
> |        LCS(1%)       |  9.94  |  19.79  |
> |      **Advanced MS**     |  1.46  |  8.94   |

---

### Official Review · Reviewer_XQdP · 2022-07-11

**Rating:** 7
**Confidence:** 4
**Soundness:** 4 excellent
**Presentation:** 4 excellent
**Contribution:** 4 excellent

**Summary:**

This paper introduces a sketching method where neural networks are used for specific components of the sketching. In particular, a neural network is used to encode stream/query items into an internal representation used to update/retrieve frequency values of items, which is followed by another neural network that decodes this information into actual frequency values. A meta-training procedure is also introduced, whose goal is to train the neural networks to handle the task at hand on a meta level, that is, not tied to a specific dataset. The method and its multiple aspects are experimentally evaluated, showing how its internal mechanisms behave under differing circumstances and improved performance over baseline methods built on Count sketch and Count-min sketch.


**Questions:**

There are two major points where I would appreciate an answer from the authors:
1. How is the method's robustness to the meta-training step? That is, how much variance is expected in the results under repeated experiments with different initial seeds?
2. Are there any theoretical results that can be leveraged from existing literature on sketches? It seems to me the proposed method is lacking on the theoretical side.

Minor points:
- l. 97: what is meant by "length"? Perhaps $l_r$ elements has less ambiguous meanings (cf. length vs. norm)
- Footnote 2 (p. 3): does that mean $a_i^\top \in \mathbb{R}^{d_2 \times d_1 \times 1}$? Please write explicitly the space where $a_i^\top$ comes from to avoid confusion
- l. 271: $\phi$ does not seem to be defined


**Limitations:**

I believe the authors have sufficiently addressed the limitations of their method by analyzing its multiple components and how they behave under different circumstances.

**Strengths And Weaknesses:**

Overall, I find this to be a strong submission. The paper is properly motivated, well-organized, and clearly written. The introduced methodology is quite original for sketching, leveraging neural networks and increasing the robustness to changes in distribution significantly, compared to existing approaches, without additional training.

I was also pleased with the experimental evaluation, as it not only compares accuracy in frequency estimates under varied conditions, but also evaluates the different components that comprise the approach. In my view, this helps justifying why certain parts are a positive contributor to the overall performance of the method, shedding also some light on when they might fail. One weakness, however pointed out by the authors, is that the proposed method needs to be extended to other sketching tasks supported by traditional sketches.

---

> ### Author Response · Authors · 2022-08-02
> **Response to Reviewer XQdP**
>
> Thanks a lot for your constructive comments! We respond to your comments below.
>
> >### Q1: How is the method's robustness to the meta-training step...
>
> Thanks. In our opinion, the robustness of the meta-training step depends on the randomness of initial seeds selection. To test that, we randomly select 4 batches of seeds for networking initialization, and collect the results of AAE and ARE and their variances, for both basic and advanced meta-sketches. Here, Basic MS 1-4 refer to the basic meta-sketches run under the 4 randomly selected initial seeds. After that, each basic meta-sketch is adapted under 4 newly initialized seeds, so as to generate 16 advanced meta-sketches, represented by Advanced MS $i$_$j$. Here, $i$ denotes the random seeds in the pre-training phase, and $j$ denotes the random seeds in the adaptation phase.
>
>
> |     MODEL SEED     |   ARE  |   AAE  |
> |:------------------:|:------:|:------:|
> |     Basic MS 1     | 10.35  | 28.98  |
> |     Basic MS 2     | 10.93  | 30.28  |
> |     Basic MS 3     |  9.79  | 29.44  |
> |     Basic MS 4     |  8.85  | 27.08  |
> |   **Variance 1 - 4**   |  **0.79**  |  **1.84**  |
> |   Advanced MS 1_1  |  4.03  | 22.44  |
> |   Advanced MS 1_2  |  3.63  | 23.04  |
> |   Advanced MS 1_3  |  3.50  | 22.58  |
> |   Advanced MS 1_4  |  2.98  | 22.68  |
> |   Advanced MS 2_1  |  3.72  | 22.16  |
> |   Advanced MS 2_2  |  3.63  | 22.11  |
> |   Advanced MS 2_3  |  3.10  | 21.71  |
> |   Advanced MS 2_4  |  2.83  | 21.45  |
> |   Advanced MS 3_1  |  3.15  | 21.22  |
> |   Advanced MS 3_2  |  3.69  | 22.27  |
> |   Advanced MS 3_3  |  3.63  | 21.49  |
> |   Advanced MS 3_4  |  2.98  | 23.13  |
> |   Advanced MS 4_1  |  3.20  | 20.88  |
> |   Advanced MS 4_2  |  3.59  | 21.08  |
> |   Advanced MS 4_3  |  2.87  | 22.06  |
> |   Advanced MS 4_4  |  2.55  | 20.60  |
> | **Variance 1_1 - 4_4** |  **0.17**  |  **0.58**  |
>
>
> The results show that, for both basic and advanced meta-sketches, the variance of AAE and ARE is low, demonstrating good robustness. Also, the AAE and ARE variance of the basic meta-sketch is a bit higher than that of the advanced meta-sketch. Because the basic meta-sketch is trained based on a Zipf distribution over a certain range of parameters for retaining the flexibility in being adapting to various skewed distributions. On the other hand, the advance meta-sketch focuses on specific data distributions.
>
>  >### Q2: Are there any theoretical results that can be leveraged from existing literature on sketches...
>
> Thanks for your comments. The meta-sketch is a neural structure based on memory networks. It seems to us that there is no solid theoretical analysis currently applicable for a memory network. Probably for the same reason, another neural data structure, NBF [23], is also without theoretical analysis.
>
> In existing literatures of sketches, it shows that the estimation error bound is not tight enough for practical use. For example, a CM-sketch of $d$ rows and $w$ columns guarantees that the counting error for a stream item is at most [10] $\frac{2N}{w}$ with probability at least $1-\delta$, where $d=In(\frac{1}{\delta})$ and $N$ is the the number of items of a data stream. For Word-query dataset ($n$=5000,$N$=273894) with storage budget $B$=9kB, if setting $\delta=0.001$, $d \approx 7$, and $w \approx 330$, the error bound of the CM-sketch would be around 1660. However, according to our experimental results, the maximum estimation error of CM-sketch is merely 627 for 5000 items, and the average error is 38.04 (as shown in Table 1), so that there is a big gap between them. Similar observations can be drawn from other datasets, especially when $N$ is big.
>
> In the future, we will keep track of the theoretical advances of memory networks and conventional sketches and come up with more theoretical results.
>
>
>
>
> >### Minor points
>
> 1.$l_{r}$ represents the norm of the vector $r_{i}$, $l_{r}=|r_{i}|$.
>
> 2.$a_{i}^{T} \in \mathbb{R}^{d_1\times d_2\times 1}$. In Algorithm 3, we show the details about the dimensional changes for each variable, in the supporting material.
>
> 3.$\phi$ is defined in line 239, which represents the default maximum number of training steps.

---

### Official Review · Reviewer_uz3r · 2022-07-12

**Rating:** 4
**Confidence:** 5
**Soundness:** 2 fair
**Presentation:** 3 good
**Contribution:** 2 fair

**Summary:**

The paper proposed a neural model to estimate item frequencies of data streams, which is using the thought of meta-learning. And they called the model meta-sketch. It’s said that they are the first to try to solve this problem with neural data structure and gain good performances on real datasets.


**Questions:**

Apart from the above, I have question about the update rule for M, M=M+z_i a_i. Is this adding process derived by some objective? Could the authors give this rule out directly. Please explain that it is representable for all items and streams.

**Strengths And Weaknesses:**

Strengths:
The paper is well structured and the explanation is easy to understand.

Weakness:
1.	In my view, training data and evaluation data should be isolated completely. That is, there should be a certain mount streams and items haven’t happened in the meta-tasks. Therefore, sampling training set and evaluating set from the same pool is not sufficient enough to prove the generalization.
2.	The references and baselines are out of date, more researches of recent years should be covered. And it is easy to get a lot of results if you search items like “Item Frequencies of Data Streams”.
3.	One advantage of meta-learning is that it could find better params of network faster and more efficiently. So I believe the efficiency is also a good sight of this model and it should be contained in the experiment.

---

> ### Author Response · Authors · 2022-08-02
> **Response to Reviewer uz3r**
>
> Thanks for your review, we respond to your comments below.
> >### Q1: Training data and evaluation data should be isolated completely...
>
> We would like to clarify that the training and test sets are completely isolated. Essentially, both training and test sets are comprised of *data items* following some *frequency distributions*, and the two sets are independently constituted.
>
> **1.Basic meta-sketch.** For both training and test datasets, only the domain of *data items* are shared. So, none of the data items are sampled from a same dataset. We believe it is a reasonable setting (e.g., NBF [23]). More, we demonstrate that the meta-sketch has good generalization ability, in handling “unobserved” items in the training set (see Figure 4 (a)). The *frequency distributions* of the training and test sets are also different. The training set uses the Zipf distribution ($\alpha\in[0.8,1.3]$). Conversely, the frequency distribution of the test set $T_r$ is obtained from the real data, and the Zipf distributions of test sets $T_s$($\alpha=0.5$) and $T_s$($\alpha=1.5$) are also different from that of the training set. The results are shown in Table 2.
>
> **2.Advanced meta-sketch.** The setting of *data items* is the same as that of the basic meta-sketch. The *frequency distributions* of the training and test sets are obtained by sampling from different parts of the real data, so that they are isolated. More, even the distributions of the two sets are different, i.e., item-frequency correspondence varies, we show that the meta-sketch has good generalization ability in coping with the dynamic change, as shown in Figure 5.
>
> >### Q2: The references and baselines are out of date, more researches of recent years should be covered...
>
> We would like to argue that the meta-sketch is the first neural data structure for sketching data streams. For a decade, the basic sketch structure is in the form of a 2D array, following the proposal of CM-sketch and C-sketch [10-11]. Existing works are either on augmenting the array-like structure with filters etc. for item frequency estimation (e.g., Cold filter [14] (SIGMOD18) and learned sketch [15] (ICLR19)), or on extending the array-like structure to support various queries (e.g., WavingSketch(KDD20), BurstSketch(SIGMOD21), and On-Off Sketch(VLDB21)).
>
> Our vision is to replace the array-like structure with the proposed neural structure. In the paper, we consider the basic functionality of a sketch, i.e., item frequency estimation, and categorize existing works into basic sketches, augmented sketches, and learned augmented sketches, as discussed in the introduction.
>
> For basic sketches, CM-sketch [10] and C-sketch [11] are commonly accepted baselines. For augmented sketches, Cold filter [14] is the state-of-the-art. For learned augmented sketches, the learned sketch [15] is the state-of-the-art. Note that, In [14], the competitors are CM-sketch, CU-sketch, and CM-CU sketch. In [15], the competitors are CM-sketch and C-sketch.
>
> We have tried our best to select state-of-the-art methods from all the three categories as competitors. Please kindly advise if any competitor is overlooked.
>
>
> >### Q3: The efficiency is also a good sight of this model and it should be contained in the experiment...
>
> The meta-learning method used for the meta-sketch belongs to one-shot learning, which is implemented by integrating the matching networks and memory networks. Unlike few-shot learning, such as MAML and reptile, there is no need for online parameter tuning. It means that one-shot learning can be viewed as an extreme case of few-shot learning, without the overhead of parameter tuning.
>
> The basic/advanced meta-sketch can be directly deployed for stream sketching after the offline training. Similar to traditional sketch, meta-sketch’ s write/query latency is actually fixed and low. Please refer to Table 9 in the supporting material for more details.
>
>
> >### Q4: Question about the update rule for M...
>
> The adding process (“update rule”) is a common process in memory-augmented neural networks [23, 26-28], for example in [23]:
>
> >A write is performed by running the controller to obtain a write word $w$ and address $a$, and then additively writing $w$
> to $M$, weighted by the address $a$, $M_{t+1} ← M_{t} + wa^{T}$.
>
> In [27]:
>
> > Each write head also produces a length $M$ add vector $a_t$ , which is added to the memory after the erase step has been performed: $M_t(i ) ←\widetilde {M}_t(i)+w_t(i)a_t$
>
> The only objective of the process is to additively write the embedding vector $z_i$ of each item to the M according to its corresponding address $a_i$.
>
> We are not quite sure about the question of being “representable for all items and streams”. To our best understanding, there can be two possibilities: 1) whether the process is common for all streaming items; 2) whether all items can be represented by embedding vectors. The answer to the former is **Yes**. The answer to the latter is also **Yes**.

---

### Author Response · Authors · 2022-08-07
**Feedback Request**

We thank all the reviewers for their time and constructive comments. We would like to know whether our responses have addressed your concerns. Please feel free to comment if there are any further confusions.

---

### Meta-Review · Area_Chair_BkAM · 2022-09-09

**Recommendation:** Reject
**Confidence:** Certain

**Metareview:**

The paper had mixed reviews in terms of scores but if we put the strengths and the weakness together the weakness appears stronger.
Strengths: People liked the neural only method and good experimental results on two datasets.
Weakness:  Evaluations may not make sense in data stream settings, only two datasets, no theoretical motivation etc. Paper do not cite and compare recent literature on learned/adaptive learned sketches

The AC went through the paper and did find that there are certain major arguments that needs to be made before the paper is accepted.
1. The paper is about learned sketches that typically works in a bursty environment and with a lot of distribution changes all the time. No wonder all the learning augmented method still use sketches to provide worst case guarantees and control over the estimation errors. This is more or less required if at all any learning based method claims to replace sketches. As a result, the purely neural approach which treats this as a learning problem without any theoretical understanding requires more justification and case study of real scenarios.
2. The paper did not cite and compare with several recent learned/adaptive learned sketches in the literature (including recent papers in NeurIPS/ICML)
3. Looking at supplementary martial and meta-task generation, it seems that there is a very strong assumption that distribution of frequency does not change over intervals. For instance, one of the typical use case of frequency estimation is to recover frequency in any interval of time (See papers on sketches over time or adaptive sketches) and dyadic interval tricks to extend sketches to do that. A purely learning based approach is unlikely to achieve much there.



**Award:**

No

---

### Decision · Program_Chairs · 2022-09-14

Reject